# Combined Plasma and Urinary Metabolomics Uncover Metabolic Perturbations Associated with Severe Respiratory Syncytial Viral Infection and Future Development of Asthma in Infant Patients

**DOI:** 10.3390/metabo12020178

**Published:** 2022-02-14

**Authors:** Shao Thing Teoh, Mara L. Leimanis-Laurens, Sarah S. Comstock, John W. Winters, Nikita L. Vandenbosch, Jeremy W. Prokop, André S. Bachmann, Sophia Y. Lunt, Surender Rajasekaran

**Affiliations:** 1Department of Biochemistry and Molecular Biology, Michigan State University, East Lansing, MI 48824, USA; teohshao@msu.edu; 2Department of Pediatrics and Human Development, College of Human Medicine, Michigan State University, Grand Rapids, MI 49503, USA; mara.leimanis@spectrumhealth.org (M.L.L.-L.); johnw.winters@helendevoschildrens.org (J.W.W.); prokopje@msu.edu (J.W.P.); bachma26@msu.edu (A.S.B.); 3Pediatric Intensive Care Unit, Helen DeVos Children’s Hospital, Grand Rapids, MI 49503, USA; nikita.hart@helendevoschildrens.org; 4Department of Food Science and Human Nutrition, Michigan State University, East Lansing, MI 48824, USA; comsto37@msu.edu; 5Department of Chemical Engineering and Materials Science, Michigan State University, East Lansing, MI 48824, USA; 6Office of Research, Spectrum Health, Grand Rapids, MI 49503, USA

**Keywords:** respiratory syncytial virus, pediatrics, bronchiolitis, asthma, metabolomics, liquid chromatography-mass spectrometry, partial least squares regression, critical care

## Abstract

A large percentage of infants develop viral bronchiolitis needing medical intervention and often develop further airway disease such as asthma. To characterize metabolic perturbations in acute respiratory syncytial viral (RSV) bronchiolitis, we compared metabolomic profiles of moderate and severe RSV patients versus sedation controls. RSV patients were classified as moderate or severe based on the need for invasive mechanical ventilation. Whole blood and urine samples were collected at two time points (baseline and 72 h). Plasma and urinary metabolites were extracted in cold methanol and analyzed by liquid chromatography coupled with tandem mass spectrometry (LC-MS/MS), and data from the two biofluids were combined for multivariate data analysis. Metabolite profiles were clustered according to severity, characterized by unique metabolic changes in both plasma and urine. Plasma metabolites that correlated with severity included intermediates in the sialic acid biosynthesis, while urinary metabolites included citrate as well as multiple nucleotides. Furthermore, metabolomic profiles were predictive of future development of asthma, with urinary metabolites exhibiting higher predictive power than plasma. These metabolites may offer unique insights into the pathology of RSV bronchiolitis and may be useful in identifying patients at risk for developing asthma.

## 1. Introduction

Respiratory syncytial virus (RSV) is a major cause of childhood morbidity across the world and mortality in low to middle income countries [1]. In the United States, RSV represents the most frequent cause of hospitalization in the first year of life [2]. Twenty percent of the annual birth cohort develops viral bronchiolitis that requires medical attention [3], and acutely ill patients have been reported to continue having other respiratory problems requiring medical attention throughout their childhood [4]. Improved understanding of the molecular mechanisms underlying development of severe symptoms in RSV infection is therefore crucial to identify potential risk factors for both severe RSV and other associated respiratory problems, as well as improve treatment strategies. We previously reported the results of a multidimensional RNA analysis, including gene, transcript, immune cell deconvolution, secondary infections, and immune repertoire comparison between RSV patients and controls, and discovered that infants with severe RSV had significant blood signatures connected to the immune system, interferon signaling, and cytokine activation [5]. Metabolomics has been used in a number of studies to identify associations between metabolic changes in biofluids and perturbations to the immune system; for example, altered levels of aromatic amino acids, fatty acids and energy metabolites have been reported to associate with serum inflammatory markers in a number of different medical conditions [6,7,8,9]. Here, we extend our study of the aforementioned cohort by using liquid chromatography–tandem mass spectrometry (LC-MS/MS) metabolomics to characterize metabolic perturbations associated with RSV severity in plasma and urine of these patients. LC-MS/MS has been employed in a number of metabolomic analyses of clinical samples including urinary and serum or plasma [10,11,12,13]; although less comprehensive in coverage than untargeted metabolomics, targeted LC-MS/MS has the advantage of increased sensitivity and quantitative performance [14,15], and our established method [16,17,18,19] is capable of profiling a wide range of metabolites including amino acids, phosphorylated compounds in glycolysis and pentose phosphate pathway, organic acids in the tricarboxylic acid (TCA) cycle, as well as intermediates in nucleotide biosynthesis and nucleotide sugar metabolism to detect perturbations within a variety of metabolic pathways. Since metabolomics are predictive of the development of disease [20], biomarkers identified in this study may also have prognostic utility in determining whether severe disease may develop after initial hospitalization.

An epidemiological link between bronchiolitis caused by RSV and the development of recurrent wheezing and/or asthma was first described more than 60 years ago [21]. Together with genetic predisposition to allergic sensitization (atopy), RSV infection in early life is now recognized as one of the most important risk factors for pediatric asthma [22,23,24]. Here, we also tested the hypothesis that RSV patient metabolomic profiles may be predictive for future development of asthma over the two years after pediatric ICU admission.

## 2. Results

### 2.1. Patient Demographics

A detailed description of the patient cohort can be found in our previous blood transcriptomics study [5] and are summarized in Table 1. RSV patients requiring invasive mechanical ventilation were considered severe, while those only requiring non-invasive respiratory support were considered moderate. From this patient cohort, we identified plasma and urinary metabolites correlated with severity of RSV infection, and additionally found urinary metabolites with strong predictive power for future asthma development in RSV patients.

Additional details of the patient demographics are included in Appendix A. Weight at time of study enrollment PICU admission was found to be statistically lower in RSV patients as compared to control (4.25 kg vs. 7.32 kg; *p* ≤ 0.005). It was a largely Caucasian population, and there were three premature (less than 36 weeks) infants among the RSV patients. All RSV patients presented with acute bronchiolitis, and a significant number (15/20) were also diagnosed with pneumonia; detailed diagnoses as well as the severity of illness scores for the 20 RSV patients are summarized in Appendix A. PELOD was included, given that it is a measure of the first 10 days in the ICU [25]. Both comprehensive and Day 0 PELOD scores were similar for both severe and moderate patient populations. The greatest variability was seen on day 3, whereby moderate patients had less than half the scores of the severe (2.75 vs. 7.93). Although included in the table, both Pediatric Index of Mortality 2 (PIM2) [26] and Pediatric Risk of Mortality III (PRISMIII) [27] are calculated during the first hours of ICU admission only, the time to consenting patients was within the first 24 h of admission, and their relevance is not clear in this context.

### 2.2. Principal Components Analysis Reveals Sample Clustering by RSV Severity

We quantified a total of 64 plasma and 70 urinary metabolite peaks over two time points. Unsupervised principal component analysis (PCA) was performed on the combined metabolite profiles to address sample clustering. The first two principal components (PCs) were each driven by one or two strong outliers—data points that are highly divergent from the majority of samples (Appendix B Figure A1). We therefore focused on PCs 3 and 4 and found that these two PCs cluster the samples into distinct control (non-RSV), severe RSV baseline, and severe RSV 72 h groups (Figure 1A), indicating that metabolic differences corresponding to RSV severity are reflected in the data. Control and severe RSV samples separated along PC 3, while the PC 4 axis separated baseline and 72 h samples. Most moderate RSV baseline samples fell in between the severe RSV baseline and control samples, which is consistent with these samples originating from patients exhibiting a moderate pathology.

### 2.3. Plasma Sialic Acid Metabolic Intermediates and Urinary Nucleotides Discriminate RSV Samples

To further investigate the differences between RSV and control patients, we performed PLS regression on combined plasma and urinary metabolites, using RSV severity as the response variable. Numerical values for RSV severity were assigned as follows: control, 0; moderate RSV, 1; severe RSV, 2. Since 72 h samples were not available for all RSV patients, nor control patients, we limited the analysis to baseline samples for the RSV patients. We first investigated the PLS score plot and confirmed that the model successfully identified metabolite trends that correspond to RSV severity, with control and severe RSV samples falling on opposite ends of the plot and moderate RSV samples in the middle (Figure 1B).

To identify plasma metabolites that correlate positively or negatively with RSV severity, we examined the PLS coefficients (Figure 2A, Appendix A). Metabolites with large positive coefficients were positively correlated with the RSV response variable. Plasma metabolites with the highest positive coefficients were xanthosine, phosphoserine, and N-acetylneuraminate. Box plots of the normalized metabolite levels showed that these metabolites were highest in severe RSV and lowest in control groups (xanthosine, *p* = 7.34 × 10^−6^; phosphoserine, *p* = 8.5 × 10^−5^; N-acetylneuraminate, *p* = 1.84 × 10^−5^) (Figure 2B). Meanwhile, metabolites with negative coefficients included N-acetylglucosamine-1-phosphate, glycerol 3-phosphate, alpha ketoglutarate, and CMP-N-acetylneuraminate. In the box plots, N-acetylglucosamine-1-phosphate showed a clear stepwise decrease from controls to moderate to severe RSV (control vs. moderate RSV, *p* = 0.04; control vs. severe RSV, *p* = 4.28 × 10^−6^), while glycerol 3-phosphate, alpha ketoglutarate and CMP-N-acetylneuraminate also significantly decreased between control and severe RSV samples (glycerol 3-phosphate, *p* = 0.0163; alpha ketoglutarate, *p* = 0.00603; CMP-N-acetylneuraminate, *p* = 0.0357) (Figure 2C).

Next, we examined the PLS coefficients for urinary metabolites (Figure 3A, Appendix A). Adenosine monophosphate (AMP), uridine monophosphate (UMP), xylulose-5-phosphate, and inosine monophosphate (IMP) had large positive coefficients indicating correlation with RSV severity. From boxplots of the normalized metabolite levels, we found that the positive correlation of these metabolites with RSV severity was driven by large increases in severe RSV samples (AMP, *p* = 0.02; UMP, *p* = 0.00872; xylulose-5-phosphate, *p* = 0.207; IMP, *p* = 0.0408), while the metabolite levels were not significantly different between control and moderate RSV samples (Figure 3B). Conversely, metabolites with large negative coefficients included citrate/isocitrate, guanosine diphosphate (GDP) and cytidine diphosphate (CDP). Citrate/isocitrate showed a stepwise decrease from control to moderate and further to severe RSV (control vs. moderate RSV, *p* = 0.0303; control vs. severe RSV, *p* = 0.000361), while GDP (control vs. moderate RSV, *p* = 0.0303; control vs. severe RSV, *p* = 0.00227) and CDP (control vs. moderate RSV, *p* = 0.00433; control vs. severe RSV, *p* = 0.00318) were decreased in RSV samples compared to controls but were not different between moderate and severe RSV (Figure 3C).

### 2.4. Patient Weight and Plasma Protein Levels Negatively Correlate with RSV Severity

Unsupervised and supervised multivariate analyses identified plasma and urinary metabolites altered in RSV relative to control. Using these multivariate analysis strategies also enabled investigation of the relationships between other patient data. We performed PCA on available clinical covariates including white blood cell measurements. Protein levels quantified in the biofluid samples following metabolite extraction were included as variables in this analysis, and categorical data such as gender, race, prematurity, diet, presence of ENT problems, or classification as asthmatic within a two-year follow-up period were converted to numerical variables (Appendix B Table A1).

Length on ventilator, RSV severity, and length of stay had positive loadings on PC 1 in this model, indicating that they were all strongly correlated with each other (Figure 4A, Appendix A). We noted that the future asthma classification covariate also had a positive loading, indicating that future classification as asthmatic correlated with severe RSV. Weight, plasma protein, and age had negative loadings, indicating that they were negatively correlated with RSV severity. Thus, severe RSV patients tended to have lower weight, lower levels of plasma protein, as well as younger age. These trends were confirmed from the PCA score plot, with the color coding of data points varied to visualize RSV severity (Figure 4B), weight (Figure 4C), plasma protein (Figure 4D), and asthma classification (Figure 4E). The PC 2 of this model seems to be driven by an inverse relationship between prematurity and ear nose throat (ENT) problems, as well as birth weight (Appendix B Figure A2). The relatively high positive loading of eosinophil counts on this PC also suggests a correlation between prematurity and eosinophils.

### 2.5. Urinary Metabolites Are Predictive of Future Asthma Development

As noted above, asthma classification showed a correlation with RSV severity. Only one out of 10 control patients were classified as asthmatic in the two-year follow-up period after hospital visit. A larger proportion of moderate and severe RSV patients (3 out of 5 moderate RSV, 7 out of 15 severe RSV; in total 10 out of 20 RSV patients, or 50%) were classified as asthmatic in the following two years (Figure 5A). To investigate whether metabolite profiles of samples taken at initial hospitalization is predictive of future diagnosis of asthma, we performed PLS regression on combined plasma and urinary metabolites, using asthma classification as the response variable. Since RSV infection exerts a strong effect on metabolite profiles of baseline samples (Figure 1B), we only used metabolite data from control and 72 h timepoint RSV samples for asthma prediction to minimize potential confounding effects from RSV-driven metabolic perturbations. This strategy resulted in two out of three moderate RSV patients who developed asthma being excluded from the analysis, since 72 h samples were not collected for these patients due to early discharge from intensive care. The asthma PLS model separated asthma and non-asthma samples as indicated by the score plot (Figure 5B). Since the metabolite profiles were sampled during initial hospitalization and the patient classification as asthmatic occurred at a later date, this suggests that metabolite profiles are predictive of future asthma development in infant patients. The score plot showed that non-asthma severe RSV samples, while still clustering together with non-asthma control and moderate RSV samples, tended to fall closer to the asthma group (Figure 5B) indicating a potential underlying predisposition of severe RSV patients to develop asthma. To exclude the possibility of overlap between high-contributing metabolites of the RSV and the asthma models, we plotted PLS coefficients from the RSV model versus the asthma model and determined that there was low correlation (Pearson’s *r* = 0.417) between the two sets of coefficients (Appendix B Figure A3).

Examination of the PLS coefficients in the asthma prediction model showed that many urinary metabolites had positive coefficients and therefore are positively correlated with asthma classification, including lysine, ornithine, UMP, IMP and GDP-mannose (Figure 5C, Appendix A). The metabolite with largest negative coefficient (i.e., negatively correlated with asthma classification) was also a urinary metabolite, xanthosine. Receiver–operator characteristic (ROC) curves using either top positive and negative urinary metabolites (lysine and xanthosine) or top positive and negative plasma metabolites (uracil and citrate/isocitrate) showed area under the curve (AUC) values close to 1, indicating good predictive power of both top urinary and plasma metabolites (Appendix B Figure A4). However, AUC was larger with urinary metabolites (0.956) than plasma metabolites (0.895). Thus, urinary metabolites have a higher predictive power toward future asthma development compared to plasma metabolites.

## 3. Discussion

In this paper, we report results of a metabolomic analysis of blood and urine samples from RSV patients and healthy controls, which complements a previous transcriptomics study performed on the same cohort [5]. The current study is the combined analysis of plasma and urine, which allows us to explore associations between metabolites extracted from the two biofluids in our multivariate analysis models.

We found significant metabolite differences in the plasma and urinary metabolites of RSV patients compared to non-RSV controls. Among urinary metabolites, citrate/isocitrate strongly and negatively correlated with RSV severity (Figure 3A). This result is consistent with previous studies comparing urinary metabolites between RSV patients and healthy controls [28,29]. The studies also found perturbations in the TCA cycle that included decreased succinate in urine samples of RSV patients; in agreement with this, we found lower succinate levels in RSV samples relative to controls (Appendix B; Figure A5).

Other significant perturbations to urinary metabolites appear to involve nucleotides, specifically increased monophosphate nucleotides (AMP, UMP, IMP) and decreased diphosphate nucleotides (GDP, CDP) in RSV samples. Monophosphate nucleotides are converted to diphosphate nucleotides by nucleoside–monophosphate kinases (NMPK) [30], while the opposite conversion from diphosphate to monophosphate nucleotides is mediated by nucleoside pyrophosphatase/phosphodiesterases (NPPs) [31]. Hence, accumulation of monophosphate nucleotides and decrease in diphosphate nucleotides may be due to either decreased NMPK activity, and/or increased NPP activity in urine-associated tissues such as the kidney or urinary tract.

We also observed apparent perturbations in the plasma metabolites of RSV patients compared to controls, particularly to the hexosamine and sialic acid biosynthesis pathways. N-acetylglucosamine-1-phosphate is an intermediate in hexosamine biosynthesis, which produces UDP-N-acetylglucosamine that in turn feeds into the sialic acid biosynthesis pathway. The product of sialic acid biosynthesis, N-acetylneuraminate, is further conjugated with CMP to form CMP-N-acetylneuraminate, the activated form that is used as substrate by sialyltransferases; sialyltransferase reactions attach the N-acetylneuraminate moiety onto glycan chains of glycoproteins and glycolipids destined for the cell surface. In addition to N-acetylglucosamine-1-phosphate, which had the largest negative coefficient in the RSV model (Figure 2A), both UDP-N-acetylglucosamine and CMP-N-acetylneuraminate were also decreased in RSV samples (Appendix B Figure A6). The decreased levels of these metabolites suggest a decrease in hexosamine and sialic acid pathway activities. Free N-acetylneuraminate positively correlated with RSV severity (Figure 2B). Since N-acetylneuraminate can be recycled from endocytosed cell surface proteins or taken up directly from the extracellular environment, it is possible that dysregulation in the pathway resulted in N-acetylneuraminate accumulation despite decreased biosynthetic activity. We previously reported an increase in N-acetylneuraminate and xanthosine levels in highly metastatic mouse mammary tumors relative to less metastatic tumors [17]. In the present study, we also identified a positive correlation between xanthosine and RSV severity, which echoes the results from the previous study. Free xanthosine can arise from degradation of xanthosine monophosphate (XMP), an intermediate in de novo guanosine nucleotide biosynthesis [32]. XMP is produced from IMP by inosine 5′-monophosphate dehydrogenase (IMPDH); this reaction is important for DNA synthesis particularly in rapidly proliferating cells including microbes, cancer cells and cells of the immune system [33]. Thus, an increase in xanthosine levels may reflect increased IMPDH activity following immune system activation in the RSV patients [34].

Respiratory viral infections are the most important risk factors for the onset of wheezing in infants and small children [22,35]. It remains unclear whether bronchiolitis causes chronic respiratory symptoms, or if it is a marker for children with a genetic/environmental predisposition for developing asthma. Here, we found that metabolite profiles from this patient cohort could be used to predict whether patients hospitalized for acute RSV bronchiolitis will become classified as asthmatic within a two-year follow-up period (Figure 5B). Although most patients who developed asthma within this cohort are RSV patients (Figure 5A), the metabolite coefficients in the RSV characterization and asthma prediction models do not show a high degree of correlation (Appendix B Figure A3), indicating that asthma predisposition is driven by a separate set of underlying metabolic factors independent of RSV severity. We additionally found that urinary metabolites have high predictive power for future asthma risk (Figure 5C). The initially surprising result that urinary metabolites have stronger predictive value than plasma could be due to the extensive contact of urine with tubule epithelial cells of the kidney, which share many of the cell properties of lung epithelial cells. As urine is one of the simplest to obtain materials from patients, these correlations could have high future clinical utility, especially if these metabolites are studied in a broader non-RSV infant cohort to establish if the RSV and metabolite correlations are dependent on each other.

Symptoms of both RSV and asthma are characterized by interferon responses within the respiratory system, with a complex interaction of immune cells with lung epithelial cells [36]. As nearly every infant is exposed to RSV, but only a small percentage are hospitalized, the priming of the innate immune system by other factors is likely responsible for the severe phenotype in many cases. This same priming is likely shared in asthma. Culture experiments have shown that Th1 or Th2 cytokine priming can overstimulate the immune system and independently increase risks for both severe RSV responses and asthma development [37]. Therefore, it has been suggested that asthma development is likely independent of RSV infections. The notion of a shared underlying cause of immune priming but distinct pathways of diseases development for severe RSV and asthma is reflected in our study results, which showed that patients hospitalized for severe RSV have a higher risk of asthma development (Figure 5A) but metabolic perturbations due to RSV and metabolic signatures predictive of asthma are distinct (Appendix B Figure A4).

One caveat of the current study is that we performed relative quantification of metabolites. We identified metabolites perturbed in RSV infection as well as metabolites predictive of asthma development; our data in the present state do not allow for establishment of concentration thresholds to be used as biomarkers for RSV diagnosis or asthma prediction. Absolute quantification of metabolite concentrations in biofluids require labeled internal standards for compounds of interest, which were not available for this study. As a future direction, we propose validating the results of this study in a separate cohort and then performing absolute quantification of the metabolites of interest to determine threshold concentrations for biomarkers.

In conclusion, we identified a perturbation in plasma metabolites in the sialic acid pathway that correlate with severity of RSV infection. Changes in urinary metabolites, especially nucleotides, also correlate with RSV severity. Additionally, we determined that urinary metabolites predict future development of asthma. These results will inform further mechanistic studies into RSV pathology as well as development of biomarkers for predicting future asthma risk to inform early detection and intervention.

## 4. Materials and Methods

### 4.1. Study Population, Site and Sample Collection

Samples were collected under the IRB protocol 2107-049-SH/HDVCH. Samples were collected at the Helen DeVos Children’s Hospital (HDVCH), a quaternary-care, urban, pediatric hospital in Western, Michigan. In brief, study patients were admitted to the PICU with bronchiolitis, were positive for RSV by nasopharyngeal test, and were less than six months of age (excluding premature births less than 34 weeks gestation). Consent was obtained, and then blood and urine samples were collected. At least 0.5 mL of blood was collected in EDTA tubes on ice and immediately centrifuged at 4 °C to obtain plasma. Plasma samples were stored at −20 °C overnight and then at −80 °C until batch processing. Urine samples were collected under clinically sterile conditions at the bedside using a substrate-free vacutainer (BD^®^ #364992-1) according to manufacturer specifications and frozen at −20 °C for short-term storage, and −80 °C for long-term storage.

RSV patient samples were collected at two time points; within 24 h of PICU admission, and directly prior to intravenous (IV) catheter removal and discharge from the PICU/hospital (on average three-four days later), which reflected the sampling of the acute and stabilization/recovery phase of the illness. In total, 20 patients with RSV bronchiolitis consented. Patients were divided into either severe or moderate groupings based on need for invasive mechanical ventilation.

A total of 10 healthy, age-matched, sedation-control patients with normal airways and lungs, presenting for routine IV sedation had blood and urine samples collected at one time point. Patient cohort details are provided in Table 1. Patient data were collected, de-identified and managed using Research Electronic Data Capture (REDCap) tools.

### 4.2. Metabolite Extraction

Plasma and urine samples were thawed on ice, and 20 µL plasma or 50 µL urine were added to 500 µL ice-cold HPLC-grade methanol containing 1 µM piperazine-*N,N*′-bis (PIPES) and 1 µM camphorsulfonic acid as internal standards. Samples were vortexed and centrifuged at 16,100× *g* for 5 min at 4 °C to precipitate protein and other cell debris. The metabolite extracts were dried under nitrogen gas and stored at −80 °C until analysis. Protein precipitated during the extraction was dissolved in 0.2 mM KOH overnight and then quantified by BCA protein assay using Pierce BCA Protein Assay Kit (Thermo Fisher, Waltham, MA, USA).

### 4.3. Metabolite Analysis by LC-MS/MS

Dried metabolite extracts were suspended in 100 µL HPLC-grade water and centrifuged at 16,100× *g* for 5 min at 4 °C to remove any precipitate. The aqueous resuspensions were diluted 1:1 with 20 mM perfluoroheptanoic acid (PFHA) to obtain metabolite suspensions in 10 mM PFHA for amino acid analysis. Samples were analyzed by ion-pairing reverse phase chromatography using an Acquity HSS T3 column (2.1 × 100 mm, 1.8 µm, Waters) for separation and a Waters Quattro Micro triple quadrupole mass spectrometer operated in positive mode as mass analyzer. The LC parameters were as follows: autosampler temperature, 10 °C; injection volume, 10 µL; column temperature, 40 °C. The LC solvents were A: 1 mM PFHA in water and B: acetonitrile. Elution from the column was performed over 13 min with flow rate fixed at 0.3 mL/min and using the following gradient, defined in terms of acetonitrile %: *t* = 0, 0%; *t* = 1, 0%; *t* = 8, 65%; *t* = 8.01, 90%; *t* = 9, 90%; *t* = 9.01, 0%; *t* = 13, 0%. Mass spectra were acquired using positive-mode electrospray ionization operating in multiple reaction monitoring (MRM) mode. The capillary voltage was 1000 V, and cone voltage was 45 V. Nitrogen was used as cone gas and desolvation gas, with flow rates of 40 and 800 L/h, respectively. The source temperature was 120 °C, and desolvation temperature was 350 °C. Argon was used as collision gas at a manifold pressure of 2.31 × 10^−3^ mbar.

A second half of the aqueous resuspensions were diluted 1:1 with 2× tributylamine (TBA) solvent (see below) to obtain metabolite suspensions in 1× TBA solvent for analysis of other metabolites, including glycolytic intermediates, TCA cycle intermediates, nucleotides, and nucleotide sugars. These samples were analyzed by ion-pairing reverse phase chromatography using an Ascentis Express C18 column (5 cm × 2.1 mm, 2.7 µm, Sigma-Aldrich, St. Louis, MO, USA) for separation and a Waters Xevo TQ-S triple quadrupole mass spectrometer as mass analyzer. The LC parameters were as follows: autosampler temperature, 4 °C; injection volume, 5 µL; column temperature, 50 °C. The LC solvents were 10 mM TBA and 15 mM acetic acid in 97:3 water:methanol (“TBA solvent”) and methanol. Elution from the column was performed over 12 min with the following varying flow rate gradient, defined in terms of methanol %: *t* = 0, 0%, flow rate 0.4 mL/min; *t* = 1, 0%, flow rate 0.4 mL/min; *t* = 2, 20%, flow rate 0.3 mL/min; *t* = 3, 20%, flow rate 0.25 mL/min; *t* = 5, 55%, flow rate 0.15 mL/min; *t* = 8, 95%, flow rate 0.15 mL/min; *t* = 9.5, 95%, flow rate 0.15 mL/min; *t* = 10, 0%, flow rate 0.4 mL/min; *t* = 12, 0%, flow rate 0.4 mL/min. Mass spectra were acquired using negative-mode electrospray ionization operating in MRM mode. The capillary voltage was 3000 V, and cone voltage was 50 V. Nitrogen was used as cone gas and desolvation gas, with flow rates of 150 and 600 L/h, respectively. The source temperature was 150 °C, and desolvation temperature was 500 °C. Argon was used as collision gas at a manifold pressure of 4.3 × 10^−3^ mbar.

On both instruments, sample order was randomized to avoid systematic bias, and each sample was run twice as analytical replicates. MRM transitions for the two instruments are listed in Appendix A.

### 4.4. Data Processing and Analysis

Peak processing was performed using MAVEN [38]. For each plasma or urine sample, data from the Quattro Micro (amino acids) and TQ-S (other metabolites) runs were separately scaled to internal standard peak (camphorsulfonic acid) measured in the respective run, and then data from both platforms were combined. Samples were further normalized within the set (plasma or urine) using Probabilistic Quotient Normalization [39] to account for concentration differences in the biofluids. Analytical replicates were then averaged to obtain the final metabolite profile for each sample. Principal component analysis (PCA) and partial least squares (PLS) regression were performed in R using the *ropls* package [40] using default settings (unit variance scaling, no log-transformation, leave-one-out cross validation). Metabolite boxplots were plotted in R and statistical comparisons were performed using the Mann–Whitney U test.

### 4.5. Clinical Covariates

Urine and plasma protein levels were quantified by BCA assay following methanol precipitation and metabolite extraction as described above (Section 4.2).

Classification of patients as asthmatic was performed by pediatric pulmonologists at the hospital during outpatient visits in the two-year period following enrollment in this study. Asthma diagnosis was based on responsiveness to bronchodilators following episodic symptoms of airflow obstruction or airway hyperresponsiveness, as standardized pulmonary function testing such as spirometry could not be performed in patients under 5 years of age [41].

White blood cell count values, including total WBC, lymphocytes, eosinophils and neutrophils were obtained from Complete Blood Count (CBC) tests performed on a Sysmex XN 10 Automated Hematology Analyzer as part of the routine care for inpatients.

## Figures and Tables

**Figure 1 metabolites-12-00178-f001:**
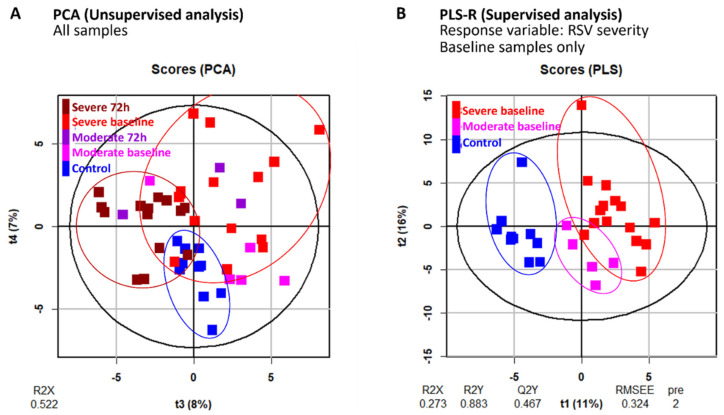
Plasma and urinary metabolite profiles reflect RSV severity. (**A**) Score plot of principal components analysis (PCA) performed using plasma and urinary metabolite data for all samples. Horizontal and vertical axes represent 3rd and 4th principal component scores, respectively. (**B**) Score plot of partial least squares (PLS) regression performed using metabolite data from control and RSV baseline samples, with RSV severity used as response variable. Horizontal and vertical axes represent 1st and 2nd PLS component scores, respectively.

**Figure 2 metabolites-12-00178-f002:**
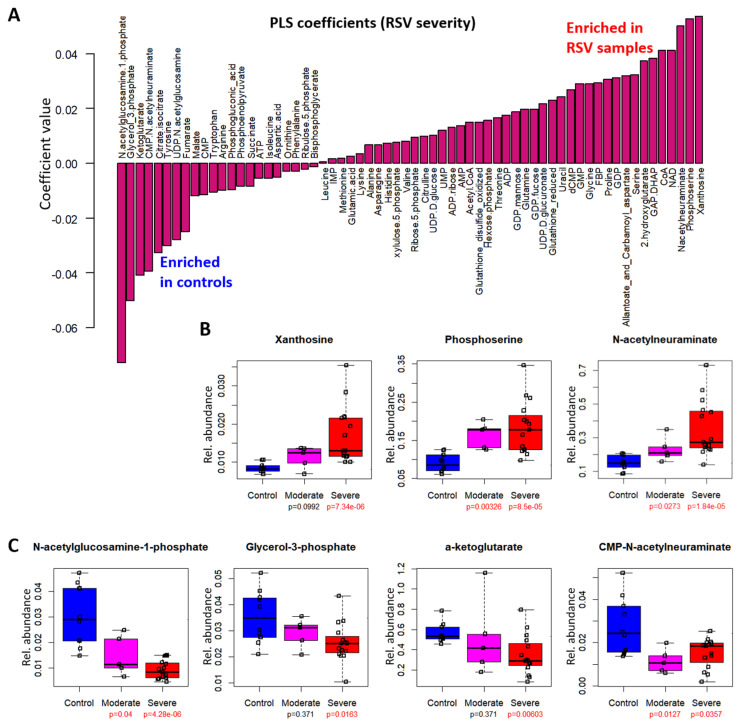
Sialic acid pathway metabolites in plasma are correlated with RSV severity. (**A**) PLS coefficients of plasma metabolites in the RSV model. Metabolites with positive coefficients are positively correlated with RSV severity, i.e., enriched in RSV samples; metabolites with negative coefficients are negatively correlated with RSV severity, i.e., enriched in controls relative to RSV samples. (**B**,**C**) Box plots of top plasma metabolites with (**B**) positive or (**C**) negative coefficients. Vertical axes represent normalized relative metabolite levels. Lower and upper ranges of the boxes represent 1st and 3rd quartiles, respectively; the center lines represent median values, and whiskers indicate maximum and minimum values. P values displayed underneath the sample class labels are calculated by the Mann–Whitney U test; significant (*p* < 0.05) values are highlighted in red.

**Figure 3 metabolites-12-00178-f003:**
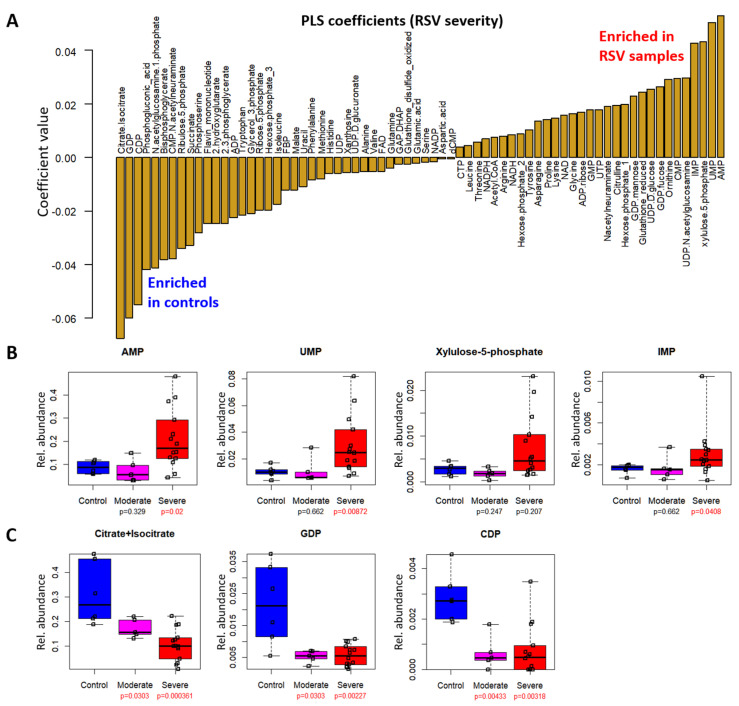
Nucleotides in urine are correlated with RSV severity. (**A**) PLS coefficients of urinary metabolites in the RSV model. Metabolites with positive coefficients are positively correlated with RSV severity, i.e., enriched in RSV samples; metabolites with negative coefficients are negatively correlated with RSV severity, i.e., enriched in controls relative to RSV samples. (**B**,**C**). Box plots of top urinary metabolites with (**B**) positive or (**C**) negative coefficients. Vertical axes represent normalized relative metabolite levels. Lower and upper ranges of the boxes represent 1st and 3rd quartiles, respectively; the center lines represent median values, and whiskers indicate maximum and minimum values. *p* values displayed underneath the sample class labels are calculated by the Mann–Whitney U test; significant (*p* < 0.05) values are highlighted in red.

**Figure 4 metabolites-12-00178-f004:**
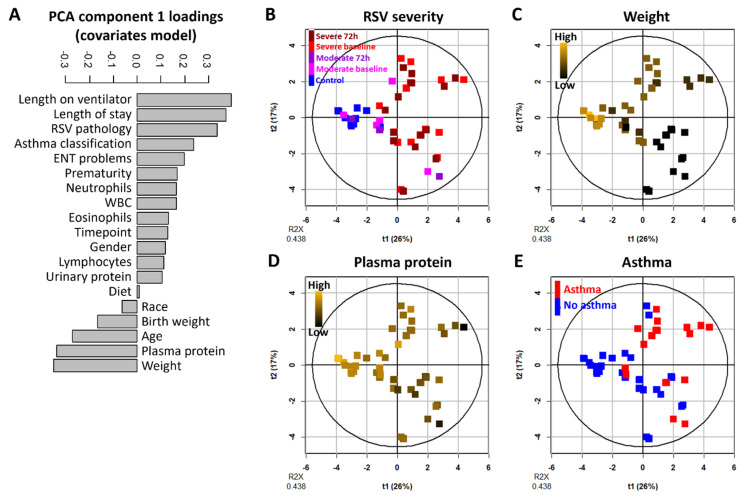
Patient weight and plasma protein are inversely correlated with RSV severity. (**A**) Covariate loadings on the 1st principal component of the covariates PCA model. Covariates with positive loadings have higher values in samples with positive 1st principal component scores, and vice versa; hence, covariates on the same side of the loading plot are positively correlated with each other, and covariates on the opposite side are inversely correlated. (**B**–**E**) Score plots all showing the 1st and 2nd principal components in the covariates-only PCA model, with sample data points color-coded according to (**B**) RSV severity, (**C**) patient weight, (**D**) plasma protein quantification, (**E**) classification of the patient as asthmatic within the two-year follow-up period after hospital visit.

**Figure 5 metabolites-12-00178-f005:**
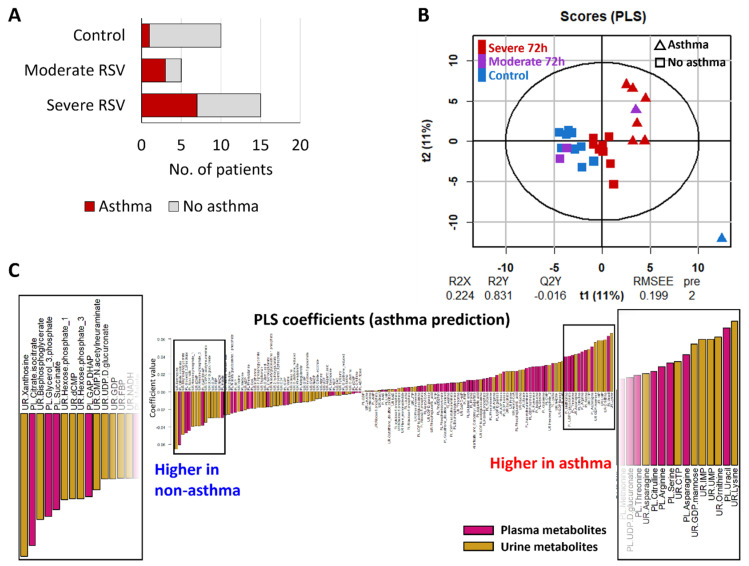
Future development of asthma is predictable from combined plasma and urinary metabolite profiles. (**A**) Stacked bar plot showing breakdown of numbers of control, moderate RSV and severe RSV patients who were diagnosed as asthmatic in the two-year follow-up period post hospital visit. (**B**) Score plot of PLS model using metabolite data from control and RSV 72 h timepoint samples, with asthma classification as response variable. Triangles represent patients classified as asthmatic while rectangles represent patients not classified as asthmatic. Horizontal and vertical axes represent 1st and 2nd PLS component scores, respectively. (**C**) PLS coefficients of metabolites in the RSV model. Metabolites with positive coefficients are positively correlated with asthma classification, i.e., enriched in patients classified as asthmatic; conversely, metabolites with negative coefficients are enriched patients not classified as asthmatic. Plasma and urinary metabolite names include “PL.” and “UR.”, respectively, and the bar plot is further color-coded (pink bars for plasma, yellow bars for urinary metabolites).

**Table 1 metabolites-12-00178-t001:** Patient cohort information for samples used in this study.

	Range	(*n*)	%	Mean	Median	St. dev.
**Age (months)**	0.5–7	30	100	2.24	2.0	1.60
Controls	0.5–6	10		3.20	3.0	1.83
RSV	0–7	20		2.00	2.0	1.52
**Gender**						
**Female**		17	56.7			
Controls		4	13.3			
RSV		13	43.3			
**Birth weight (kg)**						
Controls		10		3.65	3.7	0.40
RSV		20		3.10	3.2	0.84
**Hospital LOS ^1^ (days)**		20		14.93	11.20	9.37
Severe		15		15.73	14.58	8.63
Moderate		5		12.53	6.72	12.13
**PICU ^2^ LOS ^1^ (days)**		20		10.60	10.04	5.56
Severe		15		11.28	11.41	4.86
Moderate		5		8.56	5.53	7.58
**Respiratory Support (days)**						
Mechanical ventilation		15		8.95	7.08	4.30
Mechanical ventilation (non-invs.)		1		2.00	2.0	N/A
High flow oxygen		4		0.91	0.19	1.56

^1^ LOS: length of stay; ^2^ PICU: pediatric intensive care unit.

## Data Availability

The data presented in this study are available in article and Appendix A.

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
