# Peer review of "Combined Plasma and Urinary Metabolomics Uncover Metabolic Perturbations Associated with Severe Respiratory Syncytial Viral Infection and Future Development of Asthma in Infant Patients"

_metabolites, 2022, doi:10.3390/metabo12020178_

Round 1
Reviewer 1 Report
This paper is a very strong piece of work, and I do not have any recommendations. I think the methods section is better after the introduction, not the discussion section, but that is an editorial decision. Other than this, I recommend the publication of the paper as is.
Author Response
We thank the Reviewer for the kind and positive assessment of our work. Indeed, the placement of the Methods section after the Discussion was determined by the manuscript format which we are obliged to follow.

Reviewer 2 Report
This study by Teoh and colleagues characterizes metabolites in plasma and urine in children with respiratory syncytial viral bronchiolitis. The study compares three groups: controls, intubated children with RSV (severe), and non-intubated children with RSV (moderate) at a baseline in the hospital and for some children also at 72 hours. They describe correlates of several metabolites in plasma and, more consistently, in urine with RSV infection, severity, and to predict future asthma. While this data is interesting, there are several points that need to be addressed to better understand the characteristics of the subjects, especially for the two RSV groups and the definition of the later development of asthma in the subjects. There should also be more attention to atopy.
I suggest including more details on the study subjects at least as supplemental material instead of referring to another publication. Especially, the critical information on the characteristics (age, history of prematurity, sex, etc) of the severe and moderate subjects needs to be provided here.
The methods should include details on how the urine samples were normalized for dilution.
There need to be more details on when the baseline data was collected from the subjects, and on how sick those children were at their respective baselines.
The methods for the statement in 2.4 (lines 151-152): “We performed PCA on available clinical covariates including measurements of immune cells and protein levels in the biofluids”, need be provided, especially the type and origin of the immune cells and biofluids.
The future asthma classification that is shown in Figure 4 should be detailed in the methods.
Figure 5 B misses the data for the 3 moderate subjects who did develop asthma. This seems to be critical as this data is used to make the argument that metabolites can predict the future development of asthma.
The reference that is used to make a case for the connection of RSV and subsequent asthma is more than 60 years old and does not consider established concepts of genetic susceptibility for viral infections in children with and without atopy. This should be mentioned in the introduction.
Author Response
We thank the reviewer for the measured and helpful critique of our manuscript, particularly in pointing out inadequacies in our description of methodological and study population details. Please find our responses to the points raised in the attached document.

Reviewer 3 Report
Teoh et al. revealed that plasma and primary metabolomics, evaluated by LC-MS/MS, can help assess the severity and future asthma risk in RSV bronchiolitis infant patients. Although the number of subjects is relatively small in this study, some metabolites are expected to be an important stepping stone in identifying immune response-related mechanisms. For better scientific fidelity, please supplement the mention of the validity of LS-MS/MS and provide some explanation for the relationship of metabolomics with the serum inflammatory markers.
Author Response
We thank the Reviewer for the positive assessment of our results. We have elaborated on the validity of LC-MS/MS for our metabolomics study and included a mention of the known association of metabolomics with serum inflammatory markers in the Introduction. Please see the attached document for further details.

Round 2
Reviewer 2 Report
The revised manuscript has addressed all of the items of concern.